# Cloning and Characterization of a New β-Galactosidase from *Alteromonas* sp. QD01 and Its Potential in Synthesis of Galacto-Oligosaccharides

**DOI:** 10.3390/md18060312

**Published:** 2020-06-14

**Authors:** Dandan Li, Shangyong Li, Yanhong Wu, Mengfei Jin, Yu Zhou, Yanan Wang, Xuehong Chen, Yantao Han

**Affiliations:** School of Basic Medicine, Qingdao University, Qingdao 266071, China; 18266299974@163.com (D.L.); lisy@qdu.edu.cn (S.L.); wuyh19@126.com (Y.W.); jinmengfei678@163.com (M.J.); zy18339956716@163.com (Y.Z.); sunshine4581@163.com (Y.W.); chen-xuehong@163.com (X.C.)

**Keywords:** β-galactosidase, *Alteromonas* sp. QD01, transglycosylation, galacto-oligosaccharides

## Abstract

As prebiotics, galacto-oligosaccharides (GOSs) can improve the intestinal flora and have important applications in medicine. β-galactosidases could promote the synthesis of GOSs in lactose and catalyze the hydrolysis of lactose. In this study, a new β-galactosidase gene (*gal2A*), which belongs to the glycoside hydrolase family 2, was cloned from marine bacterium *Alteromonas* sp. QD01 and expressed in *Escherichia coli*. The molecular weight of Gal2A was 117.07 kDa. The optimal pH and temperature of Gal2A were 8.0 and 40 °C, respectively. At the same time, Gal2A showed wide pH stability in the pH range of 6.0–9.5, which is suitable for lactose hydrolysis in milk. Most metal ions promoted the activity of Gal2A, especially Mn^2+^ and Mg^2+^. Importantly, Gal2A exhibited high transglycosylation activity, which can catalyze the formation of GOS from milk and lactose. These characteristics indicated that Gal2A may be ideal for producing GOSs and lactose-reducing dairy products.

## 1. Introduction

β-galactosidase (lactase, EC3.2.1.23) is one of the most important glycoside hydrolases that can regulate the hydrolytic process of lactose by splitting the terminal nonreducing β-D-galactose components [1]. In the hydrolysis reaction, two amino acid residues participate in the catalytic cycle, with one acting as a common acid/base and the other as a nucleophile, polarizing the water molecule and targeting the glycosylase intermediate. When the same leaving group or another carbohydrate molecule receptor attacks the covalent galactosidase intermediate, a transglycosylation reaction occurs [2]. There are two main applications of β-galactosidase. One of its most important application fields is acting as a potential treatment option for lactose intolerance in addition to reducing lactose intake [3]. Lactose intolerance occurs in approximately 2/3 of the world’s population due to lactase deficiency [4]. When these individuals consume foods high in lactose, they can experience a range of symptoms, such as diarrhea, flatulence, and nausea. β-galactosidase can lessen and eradicate the symptoms through the removal of lactose to generate lactose-free milk products and conversion of lactose to glucose and galactose [5]. On the other hand, β-galactosidase can be used to synthesize galacto-oligosaccharides (GOSs) through a transglycosylation reaction. GOSs are incorporated into infant milk to reproduce valuable *bifidobacteria* in the gastrointestinal tract [6,7,8].

In recent years, prebiotics have attracted increasingly more attention because of their biological activity [9,10,11]. Prebiotics are difficult to digest directly through the stomach and intestine, but can selectively proliferate probiotics such as *lactobacillus* and *bifidobacteria* to produce short-chain fatty acids rather than harmful bacteria [12,13]. A study has showed that unsaturated alginate oligosaccharides can reduce obesity by regulating the intestinal flora [11]. As prebiotics, GOSs participate in the regulation of physiological functions and play an important role in ensuring intestinal health, immune system regulation, inhibition of tumor cell production, and protection against cardiovascular diseases [14,15,16]. In one study, Zhai et al. exhibited that GOSs can reduce lead accumulation in mice by regulating the intestinal flora and decreasing gut permeability [16]. In another study, Miyazaki et al. indicated that GOS could help maintain healthy skin by reducing the phenol produced by the intestinal flora [17]. In addition, Bouchaud et al. showed that treating pregnant mice with GOS prebiotics can reduce the offspring’s allergic reactions to food [18]. Therefore, GOSs have widely used in medical and health products because of their many desirable biological activities. However, there is a small quantity of GOSs in nature, and they are difficult to separate. Hence, hydrolyzing lactose or whey by using recombinant β-galactosidase will become the main method for obtaining a mass of GOS [19,20,21] and should be paid more attention.

Based on the similarity of their amino acid sequence, β-galactosidases have been divided into four glycoside hydrolase (GH) families, including families 1, 2, 35, and 42 (http://www.cazy.org) [22]. Some organisms can produce β-galactosidases, including bacteria, fungi, and higher eukaryotes [23]. Usually, the β-galactosidases from eukaryotes mainly belong to GH35 [24,25], while bacteria-derived β-galactosidases are mainly in GH families 42 and 2 [25,26,27,28]. The microorganisms used for the commercial production of β-galactosidase mainly include *Bifidobacterium longum*, *Bacillus subtilis*, *Lactobacillus*, *Aspergillus niger*, and *Kluyveromyces lactis* [29,30,31]. The properties and structures of β-galactosidase from different sources are quite different, so the scope of application also varies. In particular, *Kluyveromyces* is an important commercial source of β-galactosidase and is used for lactose hydrolysis and transglycosylation [32]. Moreover, the enzymes from GH2 are generally considered to be good at hydrolyzing lactose and producing oligosaccharides when compared with other GH families [33]. The analysis of the identified β-galactosidases in the GH2 family revealed that the enzymes in this family usually contain five domains, and their active sites are located in the third domain. LacZ from *Escherichia coli* is the representative of GH2 and is also the most thoroughly studied β-galactosidase [34,35]. Along with the rise of glycobiology, the study of β-galactosidase will mainly focus on improving the yield of oligosaccharides, searching for new enzyme varieties, and modifying enzyme genes.

Today, lactose intolerance and low GOS production rates are a growing concern. The primary method to solve these problems is to search for β-galactosidases with high hydrolysis and transglycosylase activity. In this study, a new β-galactosidase (Gal2A) from *Alteromonas* sp. QD01 belonging to GH2 was isolated from a Yellow Sea sample. We performed recombinant expression in *Escherichia coli* and characterized the Gal2A properties. In addition, we found that Gal2A has high transglycosylation activity and can produce GOS. Gal2A is expected to be used in medicine, food processing, and other fields.

## 2. Results and Discussion

### 2.1. Sequence Analysis of gal2A from Alteromonas sp. QD01

The marine bacteria *Alteromonas* sp. QD01 was isolated from mud samples of the Yellow Sea in China. The strain was grown in the identification medium and showed high β-galactosidase activity. Based on results of the sequence analysis, a β-galactosidase-encoding gene, *gal2A,* was found. Through the analysis of relevant websites, the *gal2A* gene contained a total of 3120 open reading frames (ORF), and the encoded a protein, Gal2A, comprised 1039 amino acids. The predicted molecular weight (Mw) and theoretical isoelectric point (pI) of its encoded β-galactosidase Gal2A were 117.07 kDa and 5.20, respectively. Further analysis revealed that Gal2A had neither signal peptides nor transmembrane domains, which confirmed that Gal2A is an intracellular enzyme. A phylogenetic tree was established by comparing the sequences of Gal2A with other β-galactosidases from the GH families 1, 35, and 42. As shown in the phylogenetic tree (Figure 1), the Gal2A showed the highest homology with a β-galactosidase from the genus *Alteromonas* sp., forming a single line cluster, followed by *Klebsiella pneumoniae* and *Escherichia coli*. According to the phylogenetic tree, the Carbohydrate-Active enZymes (CAZy) database, and the Conservative Domain Database (CDD), Gal2A was confirmed to belong to GH2.

By analyzing the Protein Data Bank (PDB) database in National Center for Biotechnology Information (NCBI), a multiple sequence alignment (Figure 2) was established between Gal2A and other three β-galactosidases reported. The three enzymes were LacZ from *Escherichia coli* (Genbank number: AAA24053), Arth βDG from *Arthrobacter* sp. 32cB (Genbank number: AHY00656), and C221-bGal from *Arthrobacter* sp. C2-2 (Genbank number: CAD29775). Based on the 3D-structure and sequence comparisons of the lacZ from *Escherichia coli* [35], the Gal2A was a large five-domain protein containing 1039 amino acids. Domain 1 (residues 48–218) and domain 2 (residues 219–334) were identified as the sugar-binding and immunoglobulin-like sandwich domain in GH2, respectively. Domain 3 (residues 335–624) was the catalytic domain containing the typical TIM-barrel with eight-stranded α/β barrel. Glu460 and Glu536 were two highly conserved glutamic acid residues in Domain 3, which act as an acid/base catalyst and a nucleophilic site, respectively [35,36,37]. Therein, Glu460 is of great significance for binding substrates and stable transition states. The function of domain 4 (residuals 625–727) is currently unknown. Domain 5 (residues 761–1030) was classified as a small chain of β-galactosidase [37,38]. Domains 1 and 5 were responsible for providing the intramolecular contacts for the request to stabilize the protein in its functional form. Furthermore, Trp567 and Trp1013 in Gal2A played a vital role in the binding of enzymes to substrates. To sum up, Gal2A had the same catalytic sites and domains as the identified β-galactosidases in GH2. This further confirmed that Gal2A belongs to GH2 and may be used for lactose hydrolysis and transglycosylation.

### 2.2. Expression and Purification of Gal2A

The gene *gal2A* was expressed in *E. coli* with the expression vector pET28a and grown in Luria Bertani (LB) medium. The Gal2A was induced by isopropyl β-D-thiogalactoside (IPTG) and purified by Ni-affinity chromatography. After the process of purification, about 120 mg of pure enzyme was obtained from 1 L of culture medium. The specific activity of purified Gal2A was 257.3 U/mg, which was significantly higher than that of the crude enzyme. As shown in Figure 3, the Mw of purified β-galactosidase Gal2A was determined to be 115 kDa by sodium dodecyl sulfate polyacrylamide gel electrophoresis (SDS-PAGE), which was similarly to the theoretical Mw (117.07 kDa).

### 2.3. Properties of Gal2A

#### 2.3.1. Effects of Temperature and pH on Activity and Stability of Gal2A

The optimal reaction temperature of Gal2A was found to be 40 °C with the maximum activity (Figure 4A). Compared with LacLM from *Lactobacillus sakei* Lb790, whose optimum temperature is 55 °C [39], and BGalH from *Halomonas* sp. S62, whose optimum temperature is 45 °C [40], the optimum reaction temperature of Gal2A was lower, which can save energy and reduce production costs (Table 1). Gal2A maintained a relatively high activity (>63%) in the temperature range of 30–45 °C, and still retained 40% of its initial activity at 20 °C. The results showed that Gal2A could catalyze substrates at low temperatures, which is conducive to suppressing the growth of bacterium with maintaining the activity of enzymes. Gal2A retained more than 50% of its maximum activity after incubation at 30 °C for 90 min (Figure 4B), and its relative activity gradually decreased with temperature increasing. Exposure to temperatures over 60 °C for more than 90 min resulted in complete inactivation of Gal2A. The optimal reaction pH of Gal2A was 8.0, and the enzyme could maintain a high activity in a pH range of 7.0–9.0 (>61%) (Figure 4C). Gal2A was stable in a pH range of 6.0–9.5 (Figure 4D), as it could maintain more than 70% of its original activity. 

In this study, Britton-Robinson buffer was used to determine the biochemical properties of Gal2A. This buffer contains borax, which may form substrate–borate complexes at high concentrations. However, when the concentration of borax is 40 mM, the complex of O4-O6 and O4-O3 with galactose is very weak [43], and at the same time, we found that no complex appeared in the measurement. In that pH range, the residual activity of Gal2A was similar to that of previously reported β-galactosidases (Table 1), such as PbBGal2A from *Paenibacillus barengoltzii* [25], GalA from *Alteromonas* sp. ANT48 [23], and recombinant Gal from *Alteromonas* sp. ML52 [27]. However, the pH stability range of Gal2A was much wider than that of BgaL [28] from *Paracoccus* sp. 32d (pH 6.0–7.0). Considering that the physiological pH of the intestine is about 6.0–8.0 [38], Gal2A may be a meaningful choice for treating lactose intolerance. Moreover, Gal2A may be a suitable enzyme for lactose hydrolysis in milk (pH 6.5–7.0), with prebiotic GOSs as hydrolysis products, which can improve intestinal flora and have important applications in medicine and food.

#### 2.3.2. Effects of Metal Ions and Organic Reagents on Activity of Gal2A

When β-galactosidases are used in different yields of industrial production, most of them are limited by adverse reaction conditions. Therefore, for the purpose of extensive applications, it was necessary to study the stability of β-galactosidases in other reaction conditions, such as organic solvents and metal ions. As shown in Table 2, the addition of metal ions Na^+^, K^+^, Ba^2+^, Co^2+^, Mg^2+^, Fe^3+^, Li^+^, Al^3+^, NH_4_^+^, and Mn^2+^ had a positive effect on Gal2A activity, with Mn^2+^ and NH_4_^+^ increasing the enzyme activity to 199.6% and 160.7%, respectively. In the case of the previously reported β-galactosidase GalA [23], only the metal ions of Mn^2+^, Mg^2+^, and Fe^3+^ could improve its activity; however, most cations could promote Gal2A activity. Surprisingly, the metal ion Mg^2+^ could strongly inhibit the activity of Gal from *Alteromonas* sp. ML52 over 68.0% [27], but the activity of Gal2A increased to 154.9%. By contrast, Cu^2+^ and Zn^2+^ could significantly inhibit Gal2A activity, while Ca^2+^ and Fe^2+^ could slightly decrease enzyme activity, with 95.7% and 95.6% activity remaining, respectively. As for the organic reagents, SDS had no significant effects on Gal2A activity, while EDTA could enhance it. This further suggested that the catalytic reaction of Gal2A may depend on the presence of organic reagents and metal ions.

### 2.4. Analysis of Hydrolysates from Milk and Lactose

In this study, the production of GOS was detected with 5 mg/mL lactose as the substrate within a certain reaction time. During the reaction, lactose was degraded into monomers. At the same time, the degradation products could be observed on the thin-layer chromatography (TLC) plate. In the first minute of the reaction, a small amount of GOS was generated. After 360 min of reaction, the reaction products GOS, glucose, and galactose were clearly displayed on the TLC plate (Figure 5A), and most of the lactose had been converted into different products. After a certain time, the yield of GOS decreased as a part of GOS was further hydrolyzed as the reaction continued. It is well known that β-galactosidases have important applications in medicine, such as alleviating lactose intolerance and promoting the production of GOS. Although various β-galactosidases have been cloned and identified, only a few of them can produce GOS by transglycosylation. Compared with other β-galactosidases that produce GOS, such as PbBGal2A from *Paenibacillus barengoltzii* [25] and GalA from *Alteromonas* sp. ANT48 [23], the TLC plates showed that Gal2A has the same high transglycosylation activity as these enzymes, which further demonstrated that Gal2A is a potential candidate for catalyzing the production of GOS.

We could safely conclude that Gal2A is relatively stable in the pH range of 6.5–7.0 by investigating its properties. Coincidentally, the pH of milk is also close to 6.5–7.0. Therefore, the hydrolysis progress of lactose in milk by Gal2A was examined. As it can be seen from the TLC plate (Figure 5B), there was a high lactose content in milk at the beginning, and this amount gradually decreased as the reaction continued, while the amount of galactose and GOS gradually increased. After 360 min of reaction, only a small portion of lactose was not resolved. So far, the strains *Aspergillus* sp. and *Kluyveromyces* sp. have been a major source of industrial β-galactosidases. However, the β-galactosidases from *Aspergillus* sp. have an optimum pH in the acidic range (2.5–5.4) and are not suitable for the hydrolysis of lactose in milk [1]. This indicated that Gal2A can be used to hydrolyze lactose in milk to produce GOS, and Gal2A may be a promising candidate for the production of lactose-free dairy products in the food industry.

## 3. Materials and Methods

### 3.1. Materials

Lactose, glucose, standard GOS, and galactose were supplied by Solarbio (Beijing, China). O-nitrophenyl-β-D-galactopyranoside (ONPG) and o-nitrophenol (ONP) were purchased from Sangon Biotech (Shanghai, China). Besides, the plasmid pET-28a (+) (Novagen, Madison, WI, USA) and the *E. coli* strain BL21 (DE3) (Tiangen Biotech, Beijing, China) were used for plasmid construction and gene expression host, respectively. The silica gel thin-layer chromatography (TLC) plates were purchased from Merck (Darmstadt, Germany). All reagents and chemicals were of analytical grade unless stated otherwise.

### 3.2. Isolation and Identification of Bacteria

The mud samples used were obtained from the surface of Yellow Sea sediment (depth 40 m, 120.13° E 35.76° N, collected in May, 2017). The identification medium (0.5% peptone, 1% yeast extract, 1% lactose, 0.4% NaCl, 2% agar, pH 7.0) coated with 0.004% x-gal was used to screen the β-galactosidase producing strain. After 5 days of incubation at 25 °C, at least 50 strains were isolated from detectable colonies. The hydrolysis and transglycosylation activity of the strains was determined after shaking samples in a flask in fermentative condition. Through multiple generations of screening, a highly active strain was obtained, and the 16S rDNA gene of the strain was amplified with 27F and 1492R primers. The BLASTn algorithm was used to blast the 16S rDNA gene sequence of the strain to obtain a closely related sequence, which was compared with its closely related sequence using MEGA 7.0. Finally, the strain was identified and named *Alteromonas* sp. QD01.

### 3.3. Sequence Analysis of gal2A

The coding sequence of the β-galactosidase gene from *Alteromonas* sp. QD01, *gal2A*, was cloned with two primers: Gal2ADNAF and Gal2ADNAR. To further analyze *gal2A*, the Open Reading Frame (ORF) finder in NCBI (https://www.ncbi.nlm.nih.gov/orffinder/) was used to distinguish the open reading frame. The SignalP 5.0 server (http://www.cbs.dtu.dk/services/SignalP/) and the TMHMN2.0 server (http://www.cbs.dtu.dk/services/TMHMM-2.0/) were utilized to analyze the signal peptides and transmembrane domains of *gal2A*, respectively. The BLASTp analysis in NCBI was used to study the protein sequence alignment. Moreover, the phylogenetic tree of Gal2A was constructed in MEGA 7.0 software using the bootstrapping neighbor-joining method. The family and domain information of Gal2A were obtained using the Conserved Domain Database (CDD) to refine the phylogenetic analysis. The ClustalX program and ESPript (http://espript.ibcp.fr/ESPript/ESPript/) were used to perform multiple alignment analyses of *gal2A*. Additionally, the theoretical isoelectric point (pI) and molecular weight (Mw) were calculated with the ProtParm tool as implemented in EXPASY (https://web.expasy.org/protparam/).

### 3.4. Expression and Purification of Gal2A

To express the β-galactosidase gene *gal2A* obtained by polymerase chain reaction (PCR) amplification in *E. coli*, two primers, Gal2ADNAF and Gal2ADNAR, were designed. The digested PCR product was then ligated into vector pET-28a (+) with recognition sites *Nco* I and *Xho* I, and introduced into *E. coli* BL21 (DE3) for gene expression. The *E. coli* BL21-pET28a-*gal2A* cells were grown in Luria-Bertani (LB) medium with 5 µg/mL kanamycin at 37 °C for 12 h. The cells were then transferred into Terrific Broth (TB) medium until the OD600 reached 0.6–0.8. The culture was subsequently induced with 0.1 mM IPTG and cultured at 20 °C for 24 h. The *E. coli* BL21-pET28a-*gal2A* cells were then collected after centrifugation, resuspended in a 20 mM phosphate buffer, and crushed using an ultrasonic cell disruptor (Ningboxinzhi Biotechnology Ltd., Ningbo, China) in ice condition. Finally, the cell lysates were centrifuged, and the supernatant was collected. The Akta150 FPLC purification system was used in the whole purification process. The collected supernatant was loaded to a Ni-affinity sepharose column (5 mL His-TrapTM HP), which was previously equilibrated. The washing buffer (500 mM NaCl, 20 mM phosphate buffer, pH 7.6) was used to remove the impurity protein. Besides, the eluting buffer (500 mM imidazole, 500 mM NaCl, 20 mM phosphate buffer, pH 7.6) was used to collect the active fractions. The BCA protein assay kit (Biyuntian, Shanghai, China) was used to test the concentration of the purified protein. Moreover, SDS-PAGE was used to analyze the Mw of the purified Gal2A.

### 3.5. Enzymatic Activity Assay

The activity of Gal2A could be reflected by the amount of ONP [23], whose measuring method was as follows. The reaction system contained 50 µL enzyme and 450 µL ONPG (10 mM), and ONPG was dissolved in 200 mM phosphate buffer. The reaction mixture was incubated at 40 °C for 10 min; then, the reaction was stopped by adding 500 µL Na_2_CO_3_ (1 M). After that, the mixture was centrifuged at 8000 rpm for 10 min. The absorbance of the product mixture was measured at 420 nm, which reflected the amount of ONP released from ONPG. One unit activity of β-galactosidase was defined as the amount of enzyme releasing 1 μmoL of ONP per minute under the described conditions.

### 3.6. Characterization of the Purified Gal2A

The optimum temperature of recombinant Gal2A was determined by incubating the reaction mixture at different temperatures (0–60 °C) in 200 mM phosphate buffer, pH 8.0. The optimal pH of recombinant Gal2A was evaluated at its optimal temperature in Britton-Robinson buffers (pH 5.5–10.3). This buffer solution was made up of phosphoric acid, boric acid, and acetic acid, and different amounts of sodium hydroxide can be added to form a buffer solution with a wide pH range. Then, the effects of temperature and pH on the stability of Gal2A were further investigated. To determine the thermostability of the enzyme, Gal2A was pre-incubated at different temperatures (0–60 °C) for 90 min. Then, the residual activity of the enzyme was determined at its optimal temperature and pH. Besides, to analysis its pH stability, Gal2A was pre-incubated in Britton-Robinson buffers (pH 5.5–10.3) at 4 °C for 12 h. The residual activity of the enzyme was then assayed under normal assay conditions. Finally, the effects of various reagents and metal ions on Gal2A activity were evaluated by assaying the enzyme in the presence of Na^+^, K^+^, Ca^2+^, Cu^2+^, Mn^2+^, Co^2+^, Fe^2+^, Fe^3+^, Mg^2+^, Zn^2+^, Al^3+^, NH_4_^+^, Li^+^, Ba^2+^, EDTA, and SDS (1 mM, pH 8.0) in 200 mM phosphate buffer. The activity measured in the absence of any reagent was defined as 100%. All tests were carried out in triplicate, and mean values are reported.

### 3.7. Thin-Layer Chromatography Analysis of the Reaction Products in Milk and Lactose

The reaction products of Gal2A with lactose and milk were determined by TLC analysis. In a nutshell, 100 µL purified Gal2A and 900 µL lactose substrate (5 mg/mL) were incubated in 200m M phosphate buffer (pH 8.0) at 40 °C for 360 min. The products were collected at different time points (0, 1, 5, 15, 30, 60, 240, and 360 min), and the enzyme was inactivated by boiling for 10 min. Then, 4 µL of the reaction product was spotted on the silica gel TLC plate, and plates were developed with a 3:2:5 (*v*/*v*/*v*) mixture of glacial acetic acid, water, and 1-butanol. Finally, the TLC plates were treated with an aerosolized mixture of ethanol and sulfuric acid (9:1) reagent after drying. Hydrolyzed products were visualized after heating at 80 °C for 30 min. The reaction products of lactose in milk (Inner Mongolia Mengniu Dairy Group Co., Ltd., Hohhot, China) were also analyzed using the above method. The diluted milk was reacted with Gal2A at 40 °C for 360 min, and the samples were taken at regular intervals. Finally, the reaction products were observed by TLC method. Glucose, galactose, GOS, and lactose were used as detection standards.

### 3.8. Nucleotide Sequence Accession Numbers

The β-galactosidase gene (*gal2A*) of strain *Alteromonas* sp. QD01 was deposited in GenBank under the accession number MH916568.

## 4. Conclusions

In this study, a new GH2 β-galactosidase (Gal2A) from marine bacterium *Alteromonas* sp. QD01 was successfully cloned and characterized. The optimal pH and temperature of Gal2A were determined to be pH 8.0 and 40 °C, respectively. Meanwhile, Gal2A showed wide pH stability in the pH range of 6.0–9.5, and most metal ions promoted the activity of Gal2A. Importantly, the recombinant β-galactosidase Gal2A can catalyze the hydrolysis of lactose and synthesize galacto-oligosaccharides efficiently. The exceptional properties of Gal2A render it a promising enzyme for the production of GOS. In order to further study Gal2A, we will analyze its three-dimensional structure and the molecular mechanism of GOS production.

## Figures and Tables

**Figure 1 marinedrugs-18-00312-f001:**
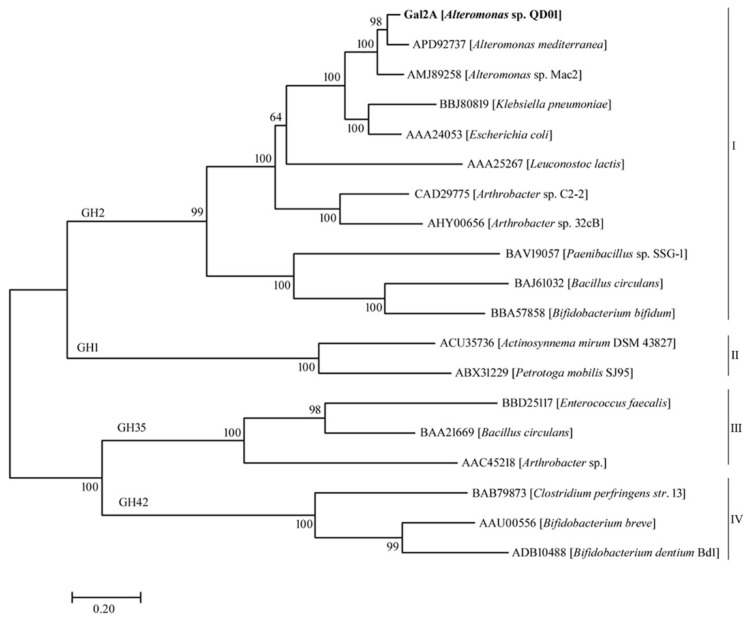
A phylogenetic tree of Gal2A and other β-galactosidases amino acid sequences based on the neighbor-joining method. The phylogenetic tree was compared using the ClustalX program and constructed using the MEGA 7.0 program.

**Figure 2 marinedrugs-18-00312-f002:**
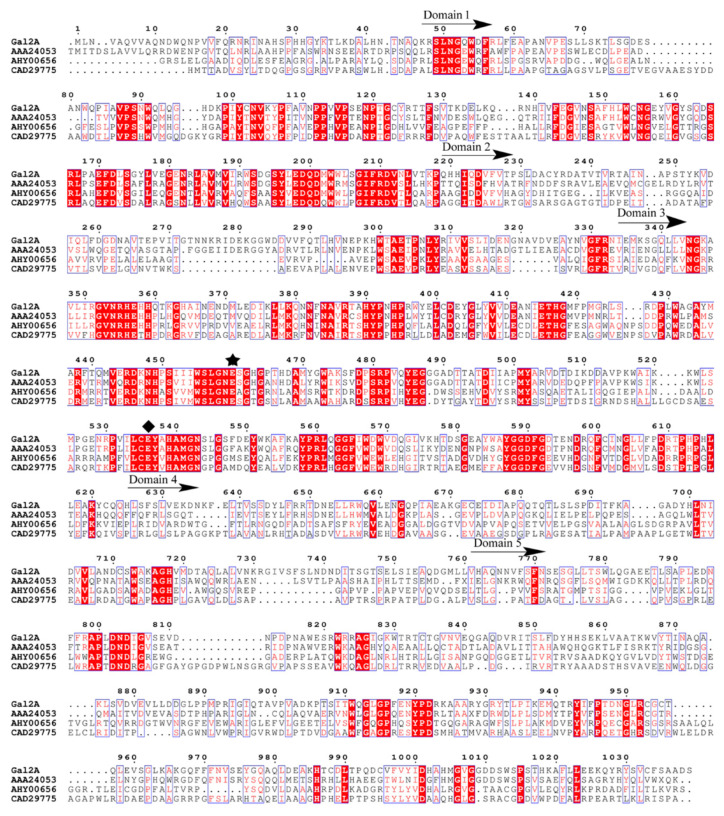
Sequence comparison of Gal2A with related β-galactosidases from glycoside hydrolase (GH) family 2: LacZ from *Escherichia coli* (Genbank number: AAA24053), ArthBDG from *Arthrobacter* sp. 32cB (Genbank number: AHY00656), and C221-bGal from *Arthrobacter* sp. C2-2 (Genbank number: CAD29775). The red background indicates an strictly conserved amino acid, and the amino acids framed by light-colored boxed are amino acid residues above a 70% consensus. The acid/base catalyst and nucleophilic sites are marked with a black pentacle and black rhombus, respectively.

**Figure 3 marinedrugs-18-00312-f003:**
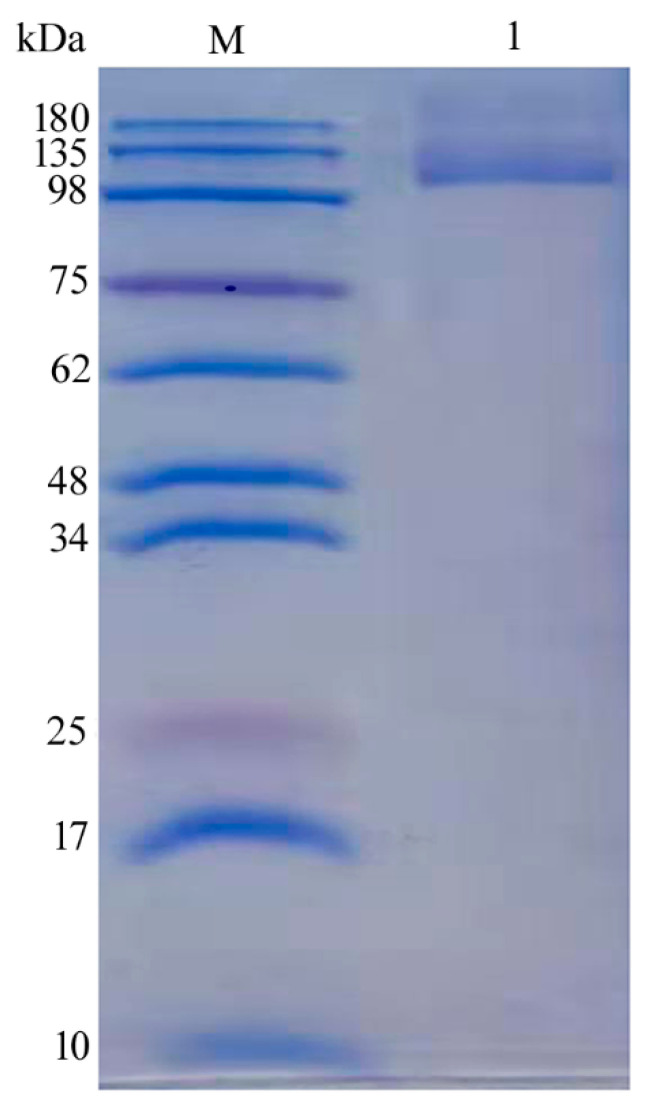
Sodium dodecyl sulfate polyacrylamide gel electrophoresis (SDS-PAGE) analysis of Gal2A. Lane M, protein marker; Lane 1, purified Gal2A.

**Figure 4 marinedrugs-18-00312-f004:**
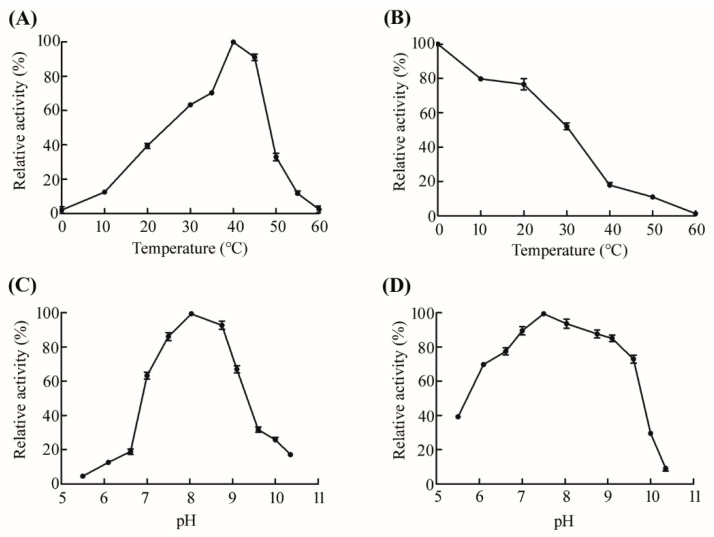
Effect of temperature and pH on Gal2A activity. (**A**) Effect of temperature on the activity of Gal2A. (**B**) Thermo-stability of Gal2A. (**C**) Effect of pH on the activity of Gal2A in Britton-Robinson buffers (pH 5.5–10.3). (**D**) pH-stability of Gal2A. Gal2A pH stability was incubated at 4 °C for 12 h in Britton-Robinson buffers (pH 5.5–10.3), and residual activity of the enzyme was then assayed under normal assay conditions. Values are the mean values ± standard deviations of three experiments in three replicates.

**Figure 5 marinedrugs-18-00312-f005:**
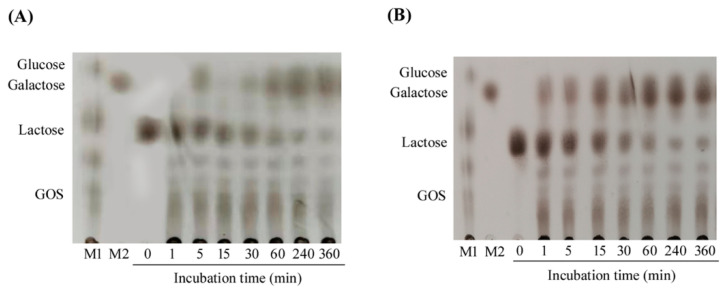
Thin-layer chromatography (TLC) analysis of Gal2A hydrolyzed lactose and milk products. (**A**) TLC analysis of Gal2A-hydrolyzed lactose. (**B**) TLC analysis of Gal2A-hydrolyzed milk. Lane M1, standard GOS; Lane M2, standard galactose.

**Table 1 marinedrugs-18-00312-t001:** Comparison of the properties of Gal2A with other β-galactosidases of GH2.

Protein Name	Source	Optimal pH	Stable pH Range	Optimal Temperature (°C)	Products	Ref.
Gal2A	*Alteromonas* sp. QD01	8.0	6.0–9.5	40	GOS, glucose, galactose	This study
Gal	*Alteromonas* sp. ML52	8.0	6.5–9.0	35	glucose, galactose	[27]
GalA	*Alteromonas* sp. ANT48	7.0	6.6–9.6	50	GOS, glucose, galactose	[23]
PbBGal2A	*Paenibacillus barengoltzii*	7.5	6.0–8.8	45	GOS, glucose, galactose	[25]
LacLM	*Lactobacillus sakei* Lb790	6.5	6.0–7.5	55	GOS, glucose, galactose	[39]
BGalH	*Halomonas* sp. S62	7.5	6.0–8.5	45	glucose, galactose	[40]
BgaL	*Paracoccus* sp. 32d	7.5	6.0–7.0	40	-	[28]
AgWH2A	*Agarivorans gilvus* WH0801	8.0	6.0–10.0	40	Agarooligosaccharides	[41]
BglA	*Arthrobacter psychrolactophilus strain* F2	8.0	6.0–10.0	10	-	[42]

**Table 2 marinedrugs-18-00312-t002:** Effects of metal ions and reagents on the activity of Gal2A.

Reagent Added	Concentration (mM)	Relative Activity (%)
None	-	100.0 ± 0.0
Na^+^	10	137.9 ± 2.6
K^+^	1	131.3 ± 3.9
Fe^2+^	1	95.6 ± 2.2
Zn^2+^	1	69.0 ± 1.1
Ba^2+^	1	130.2 ± 0.4
Co^2+^	1	128.8 ± 0.2
Mg^2+^	1	154.9 ± 5.2
Ca^2+^	1	95.7 ± 1.2
Fe^3+^	1	132.7 ± 0.9
Li^+^	1	124.1 ± 3.6
Al^3+^	1	110.6 ± 0.7
Cu^2+^	1	16.2 ± 2.2
NH_4_^+^	1	160.7 ± 3.3
Mn^2+^	1	199.6 ± 5.0
SDS	1	100.0 ± 4.2
EDTA	1	118.4 ± 2.5

Activity without addition of chemicals was defined as 100%. Data are shown as means ± SD (*n* = 3).

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
