# Peer review of "Cloning and Characterization of a New β-Galactosidase from Alteromonas sp. QD01 and Its Potential in Synthesis of Galacto-Oligosaccharides"

_marinedrugs, 2020, doi:10.3390/md18060312_

Round 1

Reviewer 1 Report

The manuscript has been revised according to my comments/suggestions. I suggesto to publish it in its present form.

Author Response

Point: The manuscript has been revised according to my comments/suggestions. I suggest to publish it in its present form.

Response: Thank you very much for your careful work and kind remind, those comments are all valuable and very helpful for revising and improving our paper, as well as the important guiding significance to our researches.

Reviewer 2 Report

The paper is now very much improved, but  the authors used Britton-Robinson buffers, which contain borax. Borax is well known to form tight complexes with carbohydrates. The enzyme stability measurements will be okay, but the pH rate profiles could be compromised by formation of substrate-borate complexes. Some stability constants are in Fig 6.4 of my Carbohydrate Chemistry and Biochemistry, and the authors could be in luck, as the O4-O6 and O4-O3 complexes with galactose would be weak enough that 40mM borate does not affect matters. Perhaps a comment should be made indicating this has been thought about, though. 

Author Response

Point: The paper is now very much improved, but the authors used Britton-Robinson buffers, which contain borax. Borax is well known to form tight complexes with carbohydrates. The enzyme stability measurements will be okay, but the pH rate profiles could be compromised by formation of substrate-borate complexes. Some stability constants are in Fig 6.4 of my Carbohydrate Chemistry and Biochemistry, and the authors could be in luck, as the O4-O6 and O4-O3 complexes with galactose would be weak enough that 40mM borate does not affect matters. Perhaps a comment should be made indicating this has been thought about, though.

Response: Thank you very much for your careful work and kind remind, this comment is very valuable and helpful for revising and improving our paper, as well as the important guiding significance to our researches.

As your suggestion, we consulted the literature and found that borax does indeed react with carbohydrates [1]. Thank you very much for your kind remind. In determining the optimal pH and pH stability of Gal2A, we used Britton-Robinson buffer, which contained 40 mM borax. However, during the entire experiment, no complex was found. Britton-Robinson buffer used to test the properties of enzymes, including β-galactosidase, is commonly used as 40 mM [2].

It is absolutely right that add some discussion is necessary. In the revised manuscript, the relevant description is as follows: In this study, Britton-Robinson buffer was used to determine the biochemical properties of Gal2A. This buffer contains borax, which may form substrate-borate complexes at high concentrations. However, when the concentration of borax is 40 mM, the complex of O4-O6 and O4-O3 with galactose is very weak [1], and at the same time we found that no complex appeared in the measurement. (Page 5, Line 153-157)

Reference

  1. Sinnott M. Carbohydrate Chemistry and Biochemistry, 1st ed.; Publisher: Royal Society of Chemistry, Britain, 2007; pp. 483–484.
  2. Sun, J.; Yao, C.; Wang, W.; Zhuang, Z.; Liu, J.; Dai, F.; Hao, J. Cloning, Expression and Characterization of a Novel Cold-adapted beta-galactosidase from the Deep-sea Bacterium Alteromonas sp. ML52. Marine drugs 2018, 16, doi:10.3390/md16120469.

Reviewer 3 Report

The paper can be accepted in the present form.

Author Response

Point: The paper can be accepted in the present form.

Response: Thank you very much for your careful work and kind remind, those comments are all valuable and very helpful for revising and improving our paper, as well as the important guiding significance to our researches.

This manuscript is a resubmission of an earlier submission. The following is a list of the peer review reports and author responses from that submission.

Round 1

Reviewer 1 Report

The manuscript reports on the isolation and characterization of a novel β-Galactosidase from Alteromonas sp. QD01 from the yellow sea and its biotechnological potentials. The manuscript is well written and methods are appropriate. Results are properly discussed. I only suggest to reduce the introduction by reporting main knowledge. Moreover, at lines 221-222, please give more details on the mud sampling, bacterial isolation and identification, and screening for  β-galactosidase activity. Add references on the methods.

Author Response

Response to Reviewer 1 Comments

Point: The manuscript reports on the isolation and characterization of a novel β-Galactosidase from Alteromonas sp. QD01 from the yellow sea and its biotechnological potentials. The manuscript is well written and methods are appropriate. Results are properly discussed. I only suggest to reduce the introduction by reporting main knowledge. Moreover, at lines 221-222, please give more details on the mud sampling, bacterial isolation and identification, and screening for β-galactosidase activity. Add references on the methods.

Response: Thank you very much for your careful work and kind remind, those comments are all valuable and very helpful for revising and improving our paper, as well as the important guiding significance to our researches. Considering the your suggestion, we have detailed the mud sampling, bacterial isolation and identification, and screening for β-galactosidase activity as the following: The mud samples used were obtained from the surface of the Yellow Sea sediment (depth 40 m, E 120.13° N 35.76°, collected in May, 2017). The identification medium (0.5% peptone, 1% yeast extract, 1% lactose, 0.4% NaCl, 2% agar, pH 7.0) coated with 0.004% x-gal was used to screened the β-galactosidase producing strain. After 5 days of incubation at 25°C, at least 50 strains were isolated from detectable colonies. The hydrolysis activity and transglycosylation activity of the strain were determined after shaking in flask fermentative condition. Through multiple generations of screening, a highly active strain was obtained, and the 16S rDNA gene of the strain was amplified with 27F and 1492R primers. BLASTn algorithm was used to blast the 16S rDNA gene sequence of the strain to obtain a closely related sequence, which was compared with its closely related sequence by MEGA 7.0. Finally, the strain was identified and named Alteromonas sp. QD01. (Page 9, Line 232-233, Line 235-242)

Reviewer 2 Report

The paper reports competent molecular biological work on a GH2 β-galactosidase from a marine bacterium, with a view to its application to the synthesis of galactooligosaccharides as dietary supplements, and some very preliminary enzymological  studies. Even the limited enzymological data presented is unsound, because of the use of "Tris" buffers. The Advances in Carbohydrate Chemistry article on Escherichia coli lacZ β-galactosidase by Wallenfels and Malhotra dating from 1962 - nearly  60 years ago - used Tris (possibly because of the low heat of ionisation of a cationic buffer, and temperature independence of measured pH), and all pH dependences had to be re-measured by Jeannine Yon's group in Orsay. In fact any molecule with a few OH groups and an NH2 group will inhibit glycosidases (paper by Alan Hayes in JCS Perkin I, around 30 years ago). The  inhibitory nature of Tris is rediscovered about once a decade.

There is no mention of any size-exclusion chromatography, which would have shown whether the enzyme was an oligomer, (like lac Z) and this experiment should be done. The enzyme was purified by Ni chromatography, so presumably was expressed with a thiol tag of some sort. Was this removed before the enzyme was characterised? If not, some artificial chimera rather than the enzyme was being studied.

The metal ion dependences seem peculiar, but the continued presence of a thiol tag would explain a lot. lacZ is a magnesium enzyme. RE Huber and I differ as to the role of Mg++, Huber thinks it is just there, I think it is an electrophile whose removal markedly slows down the hydrolysis of O-glycosides but slightly accelerates hydrolysis  of glycosyl pyridinium ions This was all disentangled in the 1990's at the latest.

The GOS products should be characterised property. Moreover, the assumption that transglycosylation is always a simple Ping-Pong reaction is unwarranted: lacZ catalyses an isomerisation of  lactose to allolactose, without the glucose coming free. Huber's group did very detailed studies of the dimeric products from lactose digestion by LacZ, published in Biochemistry around 1995.

Author Response

Response to Reviewer 2 Comments

Point 1: The paper reports competent molecular biological work on a GH2 β-galactosidase from a marine bacterium, with a view to its application to the synthesis of galactooligosaccharides as dietary supplements, and some very preliminary enzymological studies. Even the limited enzymological data presented is unsound, because of the use of "Tris" buffers. The Advances in Carbohydrate Chemistry article on Escherichia coli lacZ β-galactosidase by Wallenfels and Malhotra dating from 1962 - nearly 60 years ago - used Tris (possibly because of the low heat of ionisation of a cationic buffer, and temperature independence of measured pH), and all pH dependences had to be re-measured by Jeannine Yon's group in Orsay. In fact any molecule with a few OH groups and an NH2 group will inhibit glycosidases (paper by Alan Hayes in JCS Perkin I, around 30 years ago). The inhibitory nature of Tris is rediscovered about once a decade.

Response 1: Thank you very much for your kind remind, the comments are valuable and will greatly improve the quality of our manuscript. The reviewer is absolutely right. We carefully read the Escherichia coli lacZ β-galactosidase by Wallenfels and Malhotra in 1961 and found that Tris did have inhibitory effect. The data of our laboratory also verified this result. In the Results and Discussion, we also added this related discussion and this reference, specifically as follows: Compared with phosphate buffer and glycine-NaOH buffer, the use of Tris buffer could inhibit the activity of enzymes, which is basically consistent with the results of Wallenfels and Malhotra in 1961 [1]. (Page 6, Line 154-156)

Point 2: There is no mention of any size-exclusion chromatography, which would have shown whether the enzyme was an oligomer, (like lac Z) and this experiment should be done. The enzyme was purified by Ni chromatography, so presumably was expressed with a thiol tag of some sort. Was this removed before the enzyme was characterized? If not, some artificial chimera rather than the enzyme was being studied.

Response 2: Thank you very much for your careful work and kind remind. Actually, we performed the analysis about size-exclusion chromatography using Superdex 70 column. The results indicated that β-galactosidase (Gal2A) from Alteromonas sp. QD01 appears to be a tetrameric protein.

Point 3: The metal ion dependences seem peculiar, but the continued presence of a thiol tag would explain a lot. lacZ is a magnesium enzyme. RE Huber and I differ as to the role of Mg++, Huber thinks it is just there, I think it is an electrophile whose removal markedly slows down the hydrolysis of O-glycosides but slightly accelerates hydrolysis of glycosyl pyridinium ions. This was all disentangled in the 1990's at the latest.

Response 3: Thank you very much for your careful work and kind remind, those comments are very important guiding significance to our researches. You are absolutely right. In view of the dependence of metal ions, our conclusion supports the reviewer's conclusion: Mg2+ is an electrophile whose removal markedly slows down the hydrolysis of O-glycosides, but slightly accelerates hydrolysis of glycosyl pyridinium ions. The effect of Mg2+ on enzymes is more inclined to enhance the effect rather than the necessary effect.

Point 4: The GOS products should be characterized property. Moreover, the assumption that transglycosylation is always a simple Ping-Pong reaction is unwarranted: lacZ catalyses an isomerisation of lactose to allolactose, without the glucose coming free. Huber's group did very detailed studies of the dimeric products from lactose digestion by LacZ, published in Biochemistry around 1995.

Response 4: Thank you very much for your careful work and kind remind, those comments are all valuable and very helpful for revising and improving our paper. The reviewer is absolutely right, and we read the Huber's literature published in Biochemistry around 1995 carefully. In the Introduction, we added this related information and this reference, specifically as follows: In the hydrolysis reaction, two amino acid residues participate in the catalytic cycle, one of which acts as a common acid/base and the other acts as a nucleophile, polarizing the water molecule and targeting the glycosylase intermediate. When the same leaving group or another carbohydrate molecule receptor attacks the covalent galactosidase intermediate, a transglycosylation reaction occurs [2]. (Page 1, Line 30-36)

References

[1]. K, W.; OP, M. Galactosidases. Advances in carbohydrate chemistry 1961, 16, 239-298, doi:10.1016/s0096-5332(08)60264-7.

[2]. Richard, J.P.; Huber, R.E.; Heo, C.; Amyes, T.L.; Lin, S. Structure-reactivity relationships for beta-galactosidase (Escherichia coli, lac Z). 4. Mechanism for reaction of nucleophiles with the galactosyl-enzyme intermediates of E461G and E461Q beta-galactosidases. Biochemistry 1996, 35, 12387-12401, doi:10.1021/bi961029b.

Reviewer 3 Report

In the manuscript entitled "Cloning and Characterization of a New β-Galactosidase from Alteromonas sp. QD01 and its Potential in Synthesis of Galacto-Oligosaccharides", by Li et al. (Manuscript marinedrugs-784918), the authors describe the discovery of novel beta-galactosidase from Alteromonas sp. QD01 (Gal2A). Initially, the authors performed an in silico characterization of Gal2A, revealing that Gal2A has likely a very similar overall structure/domain organization as described before for other beta-galactosidases. Next, the authors investigated the effect of temperature, pH, and metal ions on the activity of Gal2a

To the reviewer’s point of view the current version of the manuscript is very well written and comprehensive. My major issue is the lack of novelty. The author’s could not convince me, that the here presented beta-galactosidase has so many advantageous characteristics compared to already described beta-galactosidases.

Other more specific comments/questions:

  • Page 2, line 89: I guess the phylogenetic tree was calculated based on protein sequences, hence I would write “Gal2A” instead of “gal2A”.
  • Page 4, Figure 2: Please explain in the figure caption the color-coding: red background indicates strictly conserved amino acid, …..
  • Why Figure 2 is before Figure 1 in the manuscript?
  • Figure 1 is very large. I would recommended to scale it down.
  • Figure 4: Please explain the error bars. How many replicates? How were the error bars calculated?
  • Page 9, line 239: I would rather write the sentence as the theoretical pI and molecular weight was calculated by the ProtParm tool as implemented in EXPASY. “confirmed” is misleading, since you haven’t done a comparison to experimental values.
  • Figure 5: Please indicate in the figure caption what is shown on lane “M1” and “M2”

Author Response

Response to Reviewer 3 Comments

In the manuscript entitled "Cloning and Characterization of a New β-Galactosidase from Alteromonas sp. QD01 and its Potential in Synthesis of Galacto-Oligosaccharides", by Li et al. (Manuscript marinedrugs-784918), the authors describe the discovery of novel beta-galactosidase from Alteromonas sp. QD01 (Gal2A). Initially, the authors performed an in silico characterization of Gal2A, revealing that Gal2A has likely a very similar overall structure/domain organization as described before for other beta-galactosidases. Next, the authors investigated the effect of temperature, pH, and metal ions on the activity of Gal2A.

Point 1: To the reviewer’s point of view the current version of the manuscript is very well written and comprehensive. My major issue is the lack of novelty. The author’s could not convince me, that the here presented beta-galactosidase has so many advantageous characteristics compared to already described beta-galactosidases.

Response 1: Thank you very much for your careful work and kind remind, those comments are all valuable and very helpful for revising and improving our paper, as well as the important guiding significance to our researches. In this study, a new β-galactosidase gene (gal2A) was cloned from marine bacterium Alteromonas sp. QD01 and expressed in Escherichia coli. The optimal pH and temperature of Gal2A were 8.0 and 40 °C, respectively. Compared with LacLM from Lactobacillus sakei Lb790 whose optimum temperature is 55 °C [1] and BGalH from Halomonas sp. S62 whose optimum temperature is 45 °C [2], the optimum reaction temperature of Gal2A was lower, which can save energy and reduce production cost. At the same time, Gal2A showed wide pH stability in the pH range of 6.0-9.5, which is suitable for lactose hydrolysis in milk. The pH stability range of Gal2A was much wider than that of BgaL from Paracoccus sp. 32d (pH 6.0-7.0) [3]. Most metal ions promoted the activity of Gal2A, especially Mn2+ and Mg2+. Compared with previously reported the β-galactosidase GalA [4], only the metal ions of Mn2+, Mg2+, and Fe3+ could improve it activity, while most cations can promote Gal2A activity. Surprisingly, the metal ion Mg2+ could strongly inhibit the activity of the Gal from Alteromonas sp. ML52 activity over 68.0% [5], but the activity of Gal2A increased to 154.9%. Importantly, Gal2A exhibited high transglycosylation activity, which can catalyze the formation of GOS from milk and lactose. So far, the strain Aspergillus sp. and Kluyveromyces sp. were a major source of industrial β-galactosidases. However, the β-galactosidases from Aspergillus sp. have an optimum pH at acidic range (2.5-5.4) and are not suitable for hydrolysis of lactose in milk [6]. These characteristics indicated that Gal2A may be ideal for producing GOS and lactose-reducing dairy products. Therefore, compared to the already described β-galactosidase, the β-galactosidase (Gal2A) described here has many advantages.

Point 2: Page 2, line 89: I guess the phylogenetic tree was calculated based on protein sequences, hence I would write “Gal2A” instead of “gal2A”.

Response 2: Thank you very much for your suggestion, and we have changed the “gal2A” into “Gal2A”. (Page 3, Line 95)

Point 3: Page 4, Figure 2: Please explain in the figure caption the color-coding: red background indicates strictly conserved amino acid, ….

Response 3: Thank you for your instructive suggestions, we have added “The red background indicates strictly conserved amino acid, and the amino acids framed by light-colored box indicates amino acid residues above a 70% consensus” to the legend of Figure 2. (Page 4, Line 125-127)

Point 4: Why Figure 2 is before Figure 1 in the manuscript?

Response 4: Thanks for your careful work, and we are so sorry for this mistake. We have marked Figure3 as Figure1 by mistake, and we have changed the second Figure1 to Figure3 in the manuscript. (Page 5, Line 139)

Point 5: Figure 1 is very large. I would recommended to scale it down.

Response 5: Thank you very much for reminding me about that. Due to our mistakes, we mistakenly labeled Figure3 as Figure1. We also reduced the scale of Figure 3. (Page 5, Figure 3)

Point 6: Figure 4: Please explain the error bars. How many replicates? How were the error bars calculated?

Response 6: Thank you for your careful work. we have added “Values are the mean values ± standard deviations of three experiments in three replicates” to the legend of Figure 4 according to your suggestion. (Page 6, Line 173)

Point 7: Page 9, line 239: I would rather write the sentence as the theoretical pI and molecular weight was calculated by the ProtParm tool as implemented in EXPASY. “confirmed” is misleading, since you haven’t done a comparison to experimental values.

Response 7: Thank you for your instructive suggestions, and we have changed the expressions as follows: the theoretical isoelectric point (pI) and molecular weight (Mw) were calculated by the ProtParm tool as implemented in EXPASY. (Page 9, Line 254-256)

Point 8: Figure 5: Please indicate in the figure caption what is shown on lane “M1” and “M2”.

Response 8: Thanks for your valuable suggestion, we have added “Lane M1, standard GOS; Lane M2, standard galactose” to the legend of Figure 5. At the same time, we have added the source of standard GOS to the materials. (Page 8, Line 220-221; Page 9, Line 224)

References

[1] Iqbal, S.; Nguyen, T.H.; Nguyen, H.A.; Nguyen, T.T.; Maischberger, T.; Kittl, R.; Haltrich, D. Characterization of a heterodimeric GH2 beta-galactosidase from Lactobacillus sakei Lb790 and formation of prebiotic galacto-oligosaccharides. Journal of agricultural and food chemistry 2011, 59, 3803-3811, doi:10.1021/jf103832q.

[2] Wang, G.X.; Gao, Y.; Hu, B.; Lu, X.L.; Liu, X.Y.; Jiao, B.H. A novel cold-adapted beta-galactosidase isolated from Halomonas sp. S62: gene cloning, purification and enzymatic characterization. World journal of microbiology & biotechnology 2013, 29, 1473-1480, doi:10.1007/s11274-013-1311-7.

[3] Wierzbicka-Wos, A.; Cieslinski, H.; Wanarska, M.; Kozlowska-Tylingo, K.; Hildebrandt, P.; Kur, J. A novel cold-active beta-D-galactosidase from the Paracoccus sp. 32d--gene cloning, purification and characterization. Microbial cell factories 2011, 10, 108, doi:10.1186/1475-2859-10-108.

[4] Li, S.; Zhu, X.; Xing, M. A New beta-Galactosidase from the Antarctic Bacterium Alteromonas sp. ANT48 and Its Potential in Formation of Prebiotic Galacto-Oligosaccharides. Marine drugs 2019, 17, doi:10.3390/md17110599.

[5] Sun, J.; Yao, C.; Wang, W.; Zhuang, Z.; Liu, J.; Dai, F.; Hao, J. Cloning, Expression and Characterization of a Novel Cold-adapted beta-galactosidase from the Deep-sea Bacterium Alteromonas sp. ML52. Marine drugs 2018, 16, doi:10.3390/md16120469.

[6] Oliveira, C.; Guimaraes, P.M.; Domingues, L. Recombinant microbial systems for improved beta-galactosidase production and biotechnological applications. Biotechnology advances 2011, 29, 600-609, doi:10.1016/j.biotechadv.2011.03.008.

Round 2

Reviewer 2 Report

The question of the use of inhibitory Tris buffers will not be addressed by altering the manuscript. The experiments should be re-done in a non-inhibitory buffer

Reviewer 3 Report

see below